# Edge-Guided Feature Pyramid Networks: An Edge-Guided Model for Enhanced Small Target Detection

**DOI:** 10.3390/s24237767

**Published:** 2024-12-04

**Authors:** Zimeng Liang, Hua Shen

**Affiliations:** 1National Key Laboratory of Transient Physics, Nanjing University of Science and Technology, Nanjing 210094, China; 2School of Electronic and Optical Engineering, Nanjing University of Science and Technology, Nanjing 210094, China

**Keywords:** target detection, infrared small objects, feature fusion, edge characteristics

## Abstract

Infrared small target detection technology has been widely applied in the defense sector, including applications such as precision targeting, alert systems, and naval monitoring. However, due to the small size of their targets and the extended imaging distance, accurately detecting drone targets in complex infrared environments remains a considerable challenge. Detecting drone targets accurately in complex infrared environments poses a substantial challenge. This paper introduces a novel model that integrates edge characteristics with multi-scale feature fusion, named Edge-Guided Feature Pyramid Networks (EG-FPNs). This model aims to capture deep image features while simultaneously emphasizing edge characteristics. The goal is to resolve the problem of missing target information that occurs when Feature Pyramid Networks (FPNs) perform continuous down-sampling to obtain deeper semantic features. Firstly, an improved residual block structure is proposed, integrating multi-scale convolutional feature extraction and inter-channel attention mechanisms, with significant features being emphasized through channel recalibration. Then, a layered feature fusion module is introduced to strengthen the shallow details in images while fusing multi-scale image features, thereby strengthening the shallow edge features. Finally, an edge self-fusion module is proposed to enhance the model’s depiction of image features by extracting edge information and integrating it with multi-scale features. We conducted comparative experiments on multiple datasets using the proposed algorithm and existing advanced methods. The results show improvements in the IoU, nIoU, and F1 metrics, while also showcasing the lightweight nature of EG-FPNs, confirming that they are more suitable for drone detection in resource-constrained infrared scenarios.

## 1. Introduction

Infrared small object recognition is vital in the defense industry, particularly in areas like missile detection, monitoring, target tracking, and navigation [1]. Small targets are difficult to capture in infrared imagery due to their tiny size, low brightness, and weak contrast against the background [2]. Unlike conventional targets, these small objects often appear blurred in images, lacking distinct geometric shapes or texture features, making them highly susceptible to being obscured by complex backgrounds or noise, and thus significantly increasing the difficulty of detecting them [3,4,5].

Additionally, the characteristics of infrared images impose greater demands on small target recognition. Infrared imaging, which depends on thermal radiation, performs well in all weather conditions, especially during nighttime or harsh weather, and is more robust than visible light imaging. However, the low contrast, low resolution, and severe noise interference in infrared images often result in targets appearing as weak signals, making it a challenge to distinguish them from the background. Moreover, the complexity of the backgrounds is heavily influenced by atmospheric absorption, scattering, and sensor noise, which further complicate the detection [6].

Current object detection techniques are generally divided into conventional model-driven methods and data-driven approaches using deep learning. Model-driven techniques focus on leveraging physical models and background characteristics to create mathematical models for detection, emphasizing the target’s and background’s physical properties within the imaging process. These methods require a solid theoretical basis to explain the physical phenomena involved in detection and typically include filtering methods [7], local contrast enhancement [8], and low-rank decomposition [9]. Filter-based techniques, such as top-hat filtering [10], bilateral filtering [11], and two-dimensional least mean square filtering [12], aim to enhance the contrast between the target and background, but they struggle in dynamic environments and can result in over-filtering or false positives. Local contrast enhancement methods, like local mean subtraction [13] and maximum contrast patch measurement [14], also face limitations in complex backgrounds and low grayscale variations [15]. Low-rank decomposition techniques, such as the IR Patch Image (IPI) model [16] and the improved Weighted Infrared Patch Image (WIPI) model [17], perform well with stable backgrounds but lose their effectiveness in dynamic environments due to the high computational complexity [18]. Nevertheless, their reliance on specific assumptions limits their adaptability to dynamic environments, reducing their effectiveness in complex scenes.

On the other hand, data-driven methods utilize large amounts of labeled data to automatically learn detection methods using machine learning and deep learning techniques [19,20,21]. These methods take advantage of rich data resources to extract complex features from images, adapting to various scenarios and target variations. Prominent algorithms include R-CNN [22], Faster R-CNN [23], SSD [24], and YOLO [25,26,27,28,29,30,31].

However, single-scale feature extraction may have difficulty capturing targets of different sizes, especially for infrared small targets. To address this, multi-scale fusion techniques have been proposed, combining features from various scales to enhance detection across different target sizes, leading to more comprehensive and accurate results. U-net and FPN are widely adopted for multi-scale fusion, merging high-level semantic information with low-level structural details. Feature Pyramid Networks (FPNs), which combine features from multiple scales, have proven to be effective for addressing this issue. FPNs, along with their extensions like Path Aggregation Networks (PANets) and the AGPC network, enhance the model’s ability to detect targets of various sizes by integrating contextual information across different scales [32,33,34,35,36].

Moreover, incorporating edge information has further enhanced model performance. Edge information, including contours and boundaries, plays a critical role in distinguishing targets from backgrounds [37]. By combining edge detection with multi-scale fusion, deep learning models can better preserve target structure, reduce false positives and missed detections, and enhance the accuracy of infrared small object detection. Techniques that integrate edge information with deep learning models, such as RPCNet and MIFNet, have been shown to improve the precision of target localization and recognition [38,39].

However, in deep learning frameworks, repeated convolution operations may cause small infrared targets to be overlooked, reducing detection accuracy, especially in wide-field scenarios. To mitigate these issues, approaches based on segmentation can achieve pixel-level classification and localization results, helping to mitigate information loss and ensuring detection accuracy while improving operational efficiency [40]. Approaches like U-Net, UIU-Net, and ACM networks have demonstrated success in infrared target segmentation, improving boundary detection and target localization [41,42,43,44,45]. Furthermore, Generative Adversarial Networks (GANs) have been leveraged in semantic segmentation tasks for data augmentation and model optimization, helping to mitigate the foreground–background imbalance that is typical in infrared small object detection [46,47].

This paper introduces a network model that integrates edge features with multi-scale fusion, named Edge-Guided Feature Pyramid Networks (EG-FPNs). This model significantly enhances the detection accuracy for infrared small objects by effectively combining edge features and multi-scale information. EG-FPN builds upon the classical Feature Pyramid Networks (FPNs) by introducing adaptive feature extraction and multi-level feature combination modules, thus boosting the model’s ability to represent features and capture edge information. Moreover, an attention mechanism is incorporated within the fusion module to further expand the receptive field, providing a greater robustness and a higher accuracy for detecting infrared small objects in intricate backgrounds.

The detailed contributions of this paper can be summarized as follows:We designed a multi-scale feature extraction module (Adaptive Pyramid Feature Block, APF), improving the model’s ability to extract features and boosting detection accuracy by using multi-scale convolution and adaptive feature calibration mechanisms.We propose a multi-scale fusion module (Hierarchical Multi-Feature Fusion Block, HMF) that incorporates an attention mechanism to expand the receptive field for shallow semantics while assigning effective attention weights to large-scale convolution kernels, thus emphasizing edge characteristics in the model.We introduce an edge feature-based self-fusion module that integrates edge and semantic information across multiple dimensions, compensating for information loss from down-sampling and providing more accurate edge information for target detection.Compared to other cutting-edge methods, our method achieves a superior detection performance across various datasets.

## 2. Methods

### 2.1. EG-FPN Architecture Design

The EG-FPN is an infrared small object detection block built upon the FPN method. It integrates multi-scale features and dynamically adjusts feature weights at various levels to achieve efficient feature fusion. The EG-FPN architecture is illustrated in Figure 1.

Initially, an infrared image is processed by the AFP Block to extract its features. The AFP Block employs multiple layers of convolutional operations to extract multi-scale semantic information, enhancing the network’s ability to detect targets. These preliminary image features are then fed into the FPN. The FPN constructs a multi-scale feature pyramid using a top-down pathway integrated with lateral connections. This structure aids in detecting targets at multiple scales, thus enhancing the accuracy of small object detection. Within the FPN, semantic information 9e processed using the Harr Wavelet Downsampling (HWD) method. Compared to traditional downsampling methods, wavelet downsampling effectively decomposes the image to extract both low-frequency and high-frequency components, retaining more edge, texture, and detail information. After HWD, the semantic information from each layer is input into the EAR Block (Edge-Aware Residual Block). The EAR Block employs an edge self-fusion mechanism to combine edge information with semantic features, further enhancing the model’s feature representation capability. Using this foundation, the feature maps undergo processing through two additional feature fusion modules, achieving further multi-scale feature fusion. These fusion modules can adaptively adjust the weights of features across different scales, ensuring effective target detection at all scales. Finally, the detection results are output through the segmentation head. This block converts feature information into the final detection results through convolutional operations and activation functions. Algorithm 1 provides the pseudocode for the EG-FPN’s structure.
**Algorithm 1:** Our Infrared Image Processing Method**Input**: Infrared Image**begin****Do** initial convolution
  X1 = Conv(Input_Image)
**End**
**Do** Stage 1 processing
  X2 = APF(X1) 
  X3 = EAR(X2) 
  Stage1_Output = X2 + X3
**End**
**Do** Stage 2 processing
  X4 = DownSample(X2)
  X5 = APF(X4)
  X6 = EAR(X5)
  Stage2_Output = X5 + X6
**End**
**Do** Stage 3 processing
  X7 = DownSample(X5)
  X8 = APF(X7)
  X9 = EAR(X8)
  Stage3_Output = X8 + X9
**End**
**Do** Feature Fusion
  Final_Feature = Stage1_Output ⊕ Stage2_Output ⊕ Stage3_Output
**End**
**Do** Final Prediction
  result = Predict(Final_Feature)
**End**
**Output**: Binary Mask Image

Assuming that the size of the input image is 480 × 480, the output image size in each stage is shown in Table 1.

### 2.2. APF Feature Extraction Module

The Residual Block, originally proposed by Kaiming He et al. in 2015 [48], introduced skip connections, allowing information to be transmitted to the network’s deeper layers. This design alleviates the problems of vanishing and exploding gradients that typically arise as the network depth increases. The main concept behind the Residual Block is to add the input directly to the output, enabling the network to learn the residual between the input and output rather than learning the output directly, as illustrated in Figure 2.

However, traditional Residual Blocks rely on single-scale feature extraction, applying the same convolutional operations to all channels and overlooking the interrelationships between different channels. This approach may cause certain important features to be underemphasized while allowing irrelevant features to draw excessive attention. To tackle these challenges, we propose an enhanced Residual Block structure, the APF Block, which integrates multi-scale convolutional feature extraction with an inter-channel attention mechanism. The APF Block further enhances salient features through channel recalibration. The structure of the APF Block is illustrated in Figure 3.

To process an input image $I_{0}\in R^{C\times H\times W}$, convolutional layers with kernel sizes of 1 × 1, 3 × 3, 5 × 5, and 7 × 7 are initially applied to extract features at multiple scales. Each of these convolutional layers is followed by a Squeeze-and-Excitation (SE) module. This adjustment enhances significant features while diminishing less relevant ones. Consequently, this mechanism enables the model to prioritize essential features, thereby improving the efficiency of feature extraction. This process is described by Equation (1), where k∈{1×1,3×3,5×5,7×7}.
(1)$$W_{k}=\sigma (W_{2}(\mathrm{ReLU}(W_{1}(\mathrm{AvgPool}(I_{k})))))$$

In this equation, *W*_1_ and *W*_2_ are the weights of the fully connected layers, and σ represents the sigmoid function. The recalibrated feature map is denoted by $I_{k}^{*}=I_{k}⊙W_{k}$. The outputs from the various convolutional layers are concatenated to create comprehensive multi-dimensional feature information. This feature map is used to generate an attention vector through softmax, enabling the model to adjust feature weights across different scales. As a result, the final feature map can concentrate on the most critical regions of the input image. This dynamic weighting mechanism, described by Equation (2), improves the model’s detection capability in complex backgrounds.
(2)$$I_{output}=I⊙soft\max(I)$$

### 2.3. HMF Module

Existing feature fusion methods have certain limitations, such as the use of fixed fusion strategies when handling multi-scale features, the lack of modeling of relationships between features, and high computational complexity. To address the above challenges, we introduce a multi-scale feature fusion module, inspired by the ACM-FPN method. The HMF module dynamically adjusts the weights according to the input features, effectively integrating information from high-, mid-, and low-level features to achieve more targeted feature fusion. The structure of the HMF module is shown in Figure 4.

Firstly, the high-level input feature *I_h_* undergoes a series of convolution operations and activation functions, followed by an upsampling process. Similarly, the low-level input feature II undergoes analogous preprocessing steps, followed by downsampling. The details of these operations are provided in Equations (3) and (4).
(3)$$I_{h\_up}=\sigma (BN(Upsample(\mathrm{ReLU}(BN(Conv(I_{h}))))))$$
(4)$$I_{l\_down}=\sigma (BN(Downsample(\mathrm{ReLU}(BN(Conv(I_{l}))))))$$

In neural networks, high-level features usually carry more semantic information and capture global context, while low-level features focus on finer spatial details. During top-down processing, the high-level feature *I_h_up_* is refined through average pooling and batch normalization layers to reduce feature variance. Similarly, mid-level features are improved using comparable augmentation techniques. For bottom-up processing, the low-level feature *I_l_down_* utilizes the RFCAConv attention mechanism introduced by Zhang et al. [49]. The RFCAConv attention mechanism enhances feature expressiveness by combining receptive field augmentation with channel attention. The equations are as follows:(5)$$W_{h}=\sigma (BN(Conv(\mathrm{ReLU}(BN(Conv(AvgPool(I_{h\_up})))))))$$
(6)$$W_{l}=\sigma (BN(RFCAConv(\mathrm{ReLU}(BN(Conv(AvgPool(I_{l\_down})))))))$$

The features at different levels are weighted using convolutional operations, normalized via softmax, and then combined through weighted sum to produce the final feature. The process is presented in Equation (7):(7)$$I_{output}=BN(Conv(\mathrm{ReLU}(BN(Conv(I_{k})))))$$

In this equation, $I_{k}\in \{I_{h\_weighted},I_{m\_weighted},I_{l\_weighted}\}$, where $I_{h\_weighted},I_{m\_weighted}$, and $I_{l\_weighted}$ represent the high-, mid-, and low-level feature results, weighted according to their relative importance.

### 2.4. Design of the Edge-Aware Residual Block

In this paper, we propose an EAR Block, which can efficiently combine semantic and edge information, thus improving the overall feature representation capability. The design of this module is inspired by the Residual Block, and its incorporation of edge detection and fusion mechanisms helps to more effectively capture intricate details and boundary information within images. Figure 5 presents a schematic illustration of the EAR Block.

Firstly, the EAR Block extracts preliminary semantic features via a convolutional layer, then applies a non-linear transformation through the ReLU activation function. Next, the Canny edge detection algorithm is applied to capture boundary information from the image. Edge information can highlight the contours of targets and enhance feature representation, thereby reducing interference from complex backgrounds and focusing attention on the edges and shape features of the targets. This helps improve the performance of target detection. The Canny algorithm is well-known for its multi-stage edge detection process, which includes Gaussian smoothing, gradient computation, non-maximum suppression, and double-threshold detection. These methods allow the Canny algorithm to extract precise edge information. After obtaining the edge information, the EAR Block combines it with the original semantic features using a concatenation operation, leading to an enhanced feature representation. To further improve the feature expressiveness, we propose a residual connection mechanism that adds the input features directly to the fused features. This process can be described by Equation (8).
(8)H(x)=x+Canny(ReLU(Conv(x)))

## 3. Results

### 3.1. Datasets and Evaluation Metrics

There are lots of infrared small target detection datasets that are commonly used, such as NAUU-SIRST [17], NAUU-SIRST-V2 [50], MSISTD [51], and MDvsFA_cGAN [47]. The NAUU-SIRST dataset was constructed by selecting representative frames from lots of infrared tiny object sequences. NAUU-SIRST-V2 represents a significant update to NAUU-SIRST, incorporating many urban scenes, such as cranes and streetlights, and adding background interference elements. The MSISTD dataset contains 1077 images covering real and synthetic scenes, such as urban, rural, high-light, ocean, and cloud backgrounds. In addition, it includes images taken under different weather and lighting conditions, further increasing the complexity of the detection task. MDvsFA_cGAN is a large-scale open-source synthetic dataset and was created by overlaying real infrared small targets or random objects onto high-resolution natural scene images.

In this paper, we assess model performance using the IoU, nIoU, and F1 score metrics. IoU measures the ratio of the intersection area to the union area between the detected region and the ground truth. A higher IoU value reflects a greater overlap between the detection result and the actual region. The nIoU metric is a normalized IoU value. It is designed to better handle target detection tasks with varying scales and resolutions. The calculations for IoU and nIoU are illustrated in Equations (9) and (10). TP (true positive) denotes the positive samples correctly predicted by the model, FN (false negative) represents the positive samples incorrectly classified as negative, FP (false positive) refers to the negative samples wrongly classified as positive, and TN (true negative) signifies the negative samples accurately identified by the model.
(9)$$\mathrm{IoU}=\frac{\sum_{i=1}^{N}TP_{i}}{\sum_{i=1}^{N}\left(TP_{i}+FP_{i}+FN_{i}\right)}$$
(10)$$\mathrm{nIoU}=\frac{1}{N}\cdot \sum_{i=1}^{N}\frac{TP_{i}}{TP_{i}+FP_{i}+FN_{i}}$$

The F1 score balances both precision and recall and is illustrated in Equation (11):(11)$$F1=\frac{2\times precision\times recall}{precision+recall}=\frac{2TP}{2TP+FP+FN}$$

Precision is defined in Equation (12), representing the proportion of correctly predicted positive samples among all samples predicted as positive, and recall is defined in Equation (13), indicating the proportion of actual positive samples that are correctly identified by the model. A higher F1 score reflects a better balance between precision and recall.
(12)$$precision=\frac{TP}{TP+FP}$$
(13)$$recall=\frac{TP}{TP+FN}$$

Additionally, the ROC curve illustrates the dynamic relationship between the true positive rate (TPR) and the false positive rate (FPR). TPR, also known as recall, represents the fraction of actual positive samples that the model correctly classifies. Conversely, FPR indicates the proportion of negative samples that the model mistakenly classifies as positive. The corresponding equations are shown in Equations (14) and (15).
(14)$$TPR=\frac{TP}{TP+FN}$$
(15)$$FPR=\frac{FP}{FP+TN}$$

We chose IoU, nIoU, and F1 score as evaluation metrics instead of commonly used metrics like background suppression factor or signal-to-clutter ratio gain because they better address the challenges of infrared small target detection. Traditional metrics are more suited for filtering-based methods and lack meaningful evaluations for deep learning models that output binary masks. nIoU effectively handles the challenges posed by small targets with varying scales and resolutions, ensuring adaptability to diverse scenarios. Meanwhile, the F1 score provides a balanced assessment of precision and recall, making it crucial for evaluating overall model effectiveness, especially in cases of class imbalance. These metrics collectively offer a more comprehensive and accurate evaluation for infrared small target detection models.

### 3.2. Experiment Settings

The EG-FPN was built using the PyTorch 2.0.1 framework. All experiment processes were conducted on a PC equipped with an AMD Ryzen 9 5950X 3.4 GHz CPU (16 cores, 32 threads), 64 GB of RAM, and a single RTX 3090 GPU (24 GB VRAM).

To ensure fairness and effectiveness in training, validation, and testing, the datasets are divided and preprocessed with specific strategies. The NAUU-SIRST and NAUU-SIRST-V2 datasets adopt an 8:2 split ratio for training and validation. NAUU-SIRST contains 427 images, with 341 images used for training and validation and 86 for testing, while NAUU-SIRST-V2 includes 514 images, with 411 for training and validation and 103 for testing. For both datasets, representative frames are selected to ensure diversity and reduce redundancy, with NAUU-SIRST-V2 further enriching urban scene diversity. The MSISTD dataset contains 1077 images and is divided into 900 images for training and validation and 177 for testing, covering a wide range of backgrounds, weather, and lighting conditions to increase task complexity. MDvsFA_cGAN is a synthetic dataset with 1001 images, with 801 images allocated for training and validation and 200 for testing, utilizing adversarial data generation to improve robustness. The specific divisions of the experiment sets in each dataset are shown in Table 2.

Since most of the experimental networks could not leverage pretrained weights, all model architectures were trained from scratch to ensure fairness. In this work, SoftIoU [19] was chosen as the loss function, while the Nesterov Accelerated Gradient (NAG) method was employed as the optimizer. The sample threshold for calculating precision and recall was set to 0.5.

### 3.3. Comparision of the Proposed Equations with the State-of-the-Art Method

In this experiment, we compared the proposed EG-FPN method with traditional approaches (including Tophat [10], LCM [13], and MaxMedian [16]) and advanced deep learning methods (including YOLOv8 [31], UIU-Net [43], ACM [17], and AGPCNet [33]). The tests were conducted on four datasets: NAUU-SIRST, NAUU-SIRST-V2, MSISTD, and MDvsFA_cGAN. The experimental results are shown in Table 3, where the best results are highlighted in red and the second-best results are highlighted in blue.

As shown in Table 3, the EG-FPN demonstrated an outstanding performance across multiple datasets. On the NAUU-SIRST and MSISTD datasets, EG-FPN achieved the best results across all three evaluation metrics. Similarly, on the NAUU-SIRST-V2 and MDvsFA_cGAN datasets, the EG-FPN showed superior performance in terms of both IoU and F1 score, indicating that the EG-FPN method delivers consistent and effective results across different datasets, highlighting its broad applicability and reliability.

Figure 6 compares the performance of different models on the ROC curves on different datasets. The results show that the ROC curve of the EG-FPN is the closest to the top-left corner, indicating a higher true positive rate and a lower false positive rate. This demonstrates that the proposed method outperforms the other approaches in terms of its performance.

Figure 7 presents the results of various methods on different infrared images. Correctly detected results are indicated in red rectangles and magnified in the top-right corner of each image. Identified false positives are indicated by blue circles, and false negatives are indicated by yellow circles. Figure 8 presents their corresponding 3D visualizations, alongside comparative analyses with the ground truth labels.

Images 1 and 2 display infrared small target images with clouds as the background under strong light, Image 3 shows an infrared image with a forest background under strong light, Image 4 shows an image with a sea–sky background under strong light, Image 5 shows an infrared small target image with an ocean background under weak light, and Image 6 shows an infrared small target image with a forest background under weak light. From Figure 7, it can be seen that the Tophat method, as a traditional method, produced multiple false alarms and could not autonomously identify the white edges caused by the lens. YOLOv8 is a commonly used deep learning-based target detection method, which performed better than the traditional methods but is not specifically designed for infrared small target images, making it prone to missing smaller targets. The ACM, AGPCNet, and UIU-Net methods, which are specifically designed for infrared small target image detection based on deep learning, show significantly better detection performances than YOLOv8, with reduced numbers of false negatives and false positives. However, as seen in Figure 7, some of the targets they identify differ significantly from the actual small target shapes. From the tests on various infrared images, it is evident that the EG-FPN model not only accurately localizes the targets but also restores their shapes and boundaries more clearly. Compared to other models, EG-FPN exhibits almost no false positives or false negatives, indicating its superior reliability and stability. The 3D visualizations reveal that the target regions detected by the EG-FPN have strong signal intensity with clean backgrounds and minimal interference. This clear signal distinction makes the EG-FPN particularly effective in infrared small target recognition, enabling it to accurately differentiate between targets and the background.

### 3.4. Ablation Study

To validate the effectiveness of the fundamental modules in the EG-FPN, we conducted ablation experiments, ensuring that all parameters except those of the tested modules remained unchanged. Using ACM-FPN as the baseline model, we evaluated the performance of the HWD (Wavelet Downsampling), APF (Feature Extraction), EAR (Edge-Aware Residual), and HMF (Feature Fusion) modules on the MSISTD dataset. HWD effectively preserves the edges and detailed features of images through wavelet downsampling, addressing the issue of information loss in traditional downsampling methods. The APF Block significantly enhances feature extraction efficiency through multi-scale convolutions and an inter-channel attention mechanism, emphasizing salient features and thereby improving the network’s target perception capability. The EAR Block utilizes the Canny edge detection algorithm to extract precise edge information and fuses it with semantic features, further enhancing detection accuracy, especially in cases where edge information is not prominent. The HMF Block achieves more targeted feature fusion through a dynamic weight adjustment mechanism, addressing the challenges in multi-scale feature fusion and improving detection performance in complex scenarios. Through the combined application of these blocks, the EG-FPN performs exceptionally well in complex background scenarios, demonstrating its strong small target detection capability. The results of the ablation experiment are presented in Table 4. It can be seen from the table that the model’s detection performance continuously improved with the incremental addition of each module. When all modules were utilized simultaneously, the model achieved its best performance.

### 3.5. Performance Analysis

GFLOPs (Giga Floating Point Operations per Second) represent the number of floating-point operations executed per second, which is used to measure the complexity of a model. FPS (Frames Per Second) represents the number of image frames processed in one second and is used to measure the inference speed of a model. During the study, we conducted performance tests using 20 images of size 256 × 256 pixels, and the results are shown in Table 5.

According to the data in Table 4, the ACM model achieved the highest FPS, indicating the fastest processing speed, but its detection performance still lags significantly behind other methods. In contrast, the EG-FPN model has the lowest GFLOPs value and an FPS second only to ACM, suggesting that it has the lowest computational complexity and relatively fast processing speed. The EG-FPN model also demonstrates an excellent real-time capability while ensuring an optimal detection performance.

To further validate the efficiency of the EG-FPN model, we conducted a detailed analysis of its three key modules. We used the torchprofile library to calculate the GFLOPs and FPS of each module, assuming an initial input image shape of 3 × 480 × 480. Table 6 displays the GFLOPs and FPS for each module.

## 4. Discussion

In this paper, we compared the proposed EG-FPN method with various methods, including traditional image processing methods (such as Tophat, LCM, MaxMedian) and deep learning-based methods (such as YOLOv8, ACM, AGPCNet, UIU-Net). The results demonstrate that traditional methods perform poorly in complex backgrounds, often missing targets or producing false detections, and they cannot autonomously identify white edges caused by the lens. This is because traditional methods rely on hand-crafted features, which are sensitive to complex backgrounds and noise. The LCM and MaxMedian methods rely on local contrast and median filtering, which have poor adaptability to weak signals and complex backgrounds. Although the deep learning-based YOLOv8 method performs better than traditional methods, it is not specifically designed for infrared small target images and tends to miss smaller targets. Additionally, YOLOv8 is a general-purpose target detection method that has not been optimized for the characteristics of infrared small targets. ACM, AGPCNet, and UIU-Net methods are specifically designed for infrared small target image detection, but they still have limitations in handling details and boundaries. This results in significant discrepancies between some detected targets and the actual small target shapes. Based on the shortcomings of these deep learning-based methods, EG-FPN makes the following improvements:Accurate target localization: The EG-FPN model not only accurately locates targets but also more clearly restores their shapes and boundaries. EG-FPN combines edge features and multi-scale feature fusion (HMF Block), enhancing the ability to handle details and boundaries.Reduced false positives and false negatives: Compared to other models, EG-FPN has almost no false positives or false negatives, demonstrating its superior reliability and stability. EG-FPN improves feature extraction and fusion efficiency through improved residual blocks and layered feature fusion modules (APF Block).Strong signal intensity and clean background: 3D visualizations show that the target regions detected by EG-FPN have a strong signal intensity, clean backgrounds, and minimal interference. The edge self-fusion module of EG-FPN (EAR Block) enhances signal intensity and reduces background noise and interference.

Multiple results indicate that the EG-FPN consistently performs exceptionally well across multiple datasets and particularly excelling in terms of detection accuracy and adaptability. The EG-FPN outperforms other methods on several evaluation metrics, and its ROC curve is closer to the top-left corner. This performance indicates a higher true positive rate and a lower false positive rate, making its detection results more reliable. During the analysis, the EG-FPN demonstrated a high computational efficiency, achieving the lowest GFLOPs and a relatively high FPS. In the ablation study, as the modules were progressively added, the detection performance of the EG-FPN continuously improved, which validates the contribution of each module to the overall model.

In summary, targets are often obscured by complex background noise and interference in complex background scenarios. This makes it difficult for traditional methods to detect them effectively. The EG-FPN combines edge awareness with multi-scale feature fusion, effectively capturing the edge and shape features of targets, thereby enabling precise target detection in complex backgrounds.

## 5. Conclusions

This paper presents a network architecture based on boundary features and multi-scale feature fusion, aiming to capture deep image features while emphasizing edge characteristics. First, we propose an improved residual block structure, the APF Block, which incorporates multi-scale convolutions and an inter-channel attention mechanism. This significantly enhances the efficiency of feature extraction and, through channel recalibration, emphasizes salient features, thereby improving the network’s target perception ability. Second, we design a multi-scale feature fusion module by using a dynamic weight adjustment mechanism, achieving a more targeted feature fusion. The rise in the ability of the RFCAConv attention mechanisms in shallow semantics further enhances feature representation, thus improving the detection performance in complex scenarios. Additionally, we propose an Edge-Aware Residual Module (EAR), which employs the Canny edge detection algorithm to extract precise edge information and fuse it with semantic features, further improving detection accuracy. By combining the above modules, the EG-FPN infrared small target detection network is constructed, incorporating the wavelet downsampling method to effectively preserve image edges, texture information, and detailed features. Our model can efficiently detect targets across different scales. We compared our method with existing advanced methods on multiple datasets, and the experiments showed improvements in the IoU, nIoU, and F1 metrics, confirming the reliability of the proposed algorithm. And, our model is significantly lighter than the similarly effective model.

## Figures and Tables

**Figure 1 sensors-24-07767-f001:**
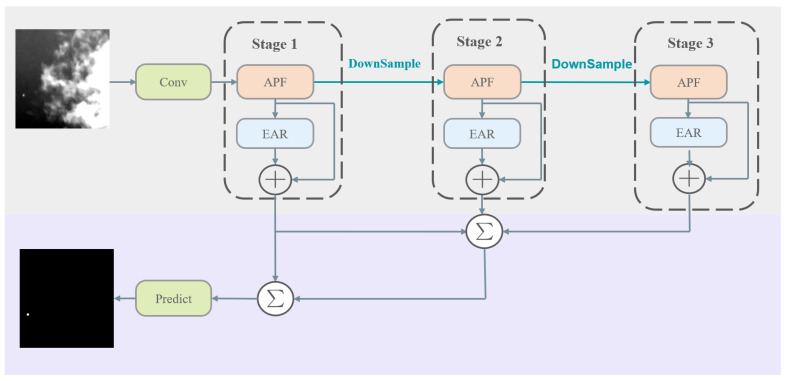
Structure of the Edge-Guided Feature Pyramid Network (EG-FPN).

**Figure 2 sensors-24-07767-f002:**
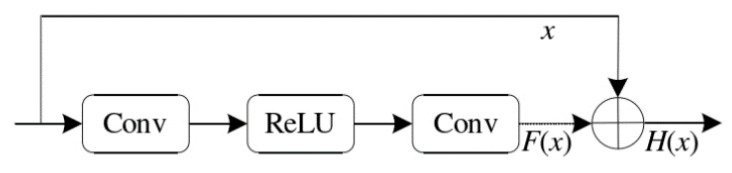
Structure of the Residual Block.

**Figure 3 sensors-24-07767-f003:**
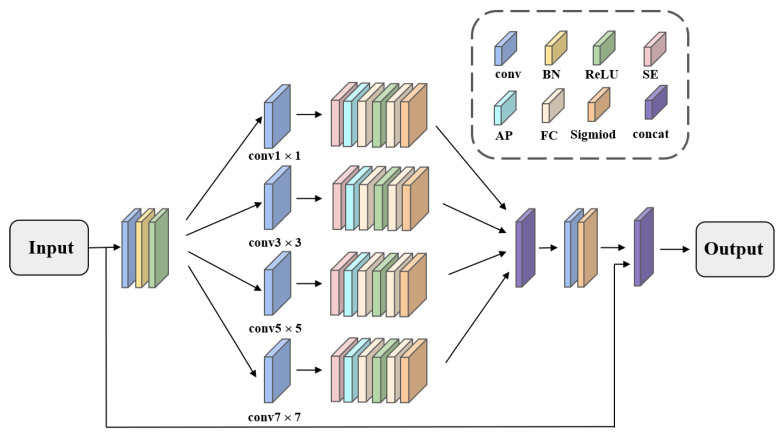
Structure of Adaptive Pyramid Feature Block (APF Block).

**Figure 4 sensors-24-07767-f004:**
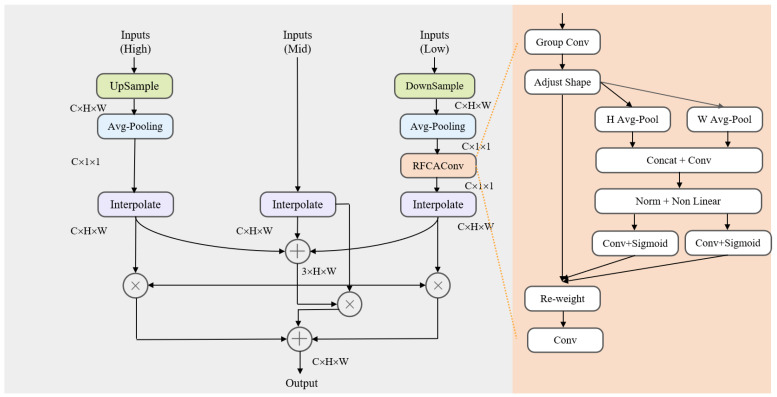
Structure of Hierarchical Multi-Feature Fusion Block (HMF Block).

**Figure 5 sensors-24-07767-f005:**
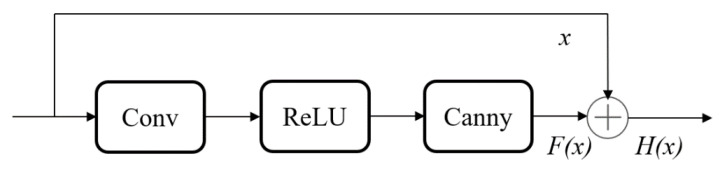
Structure of Edge-Aware Residual Block (EAR Block).

**Figure 6 sensors-24-07767-f006:**
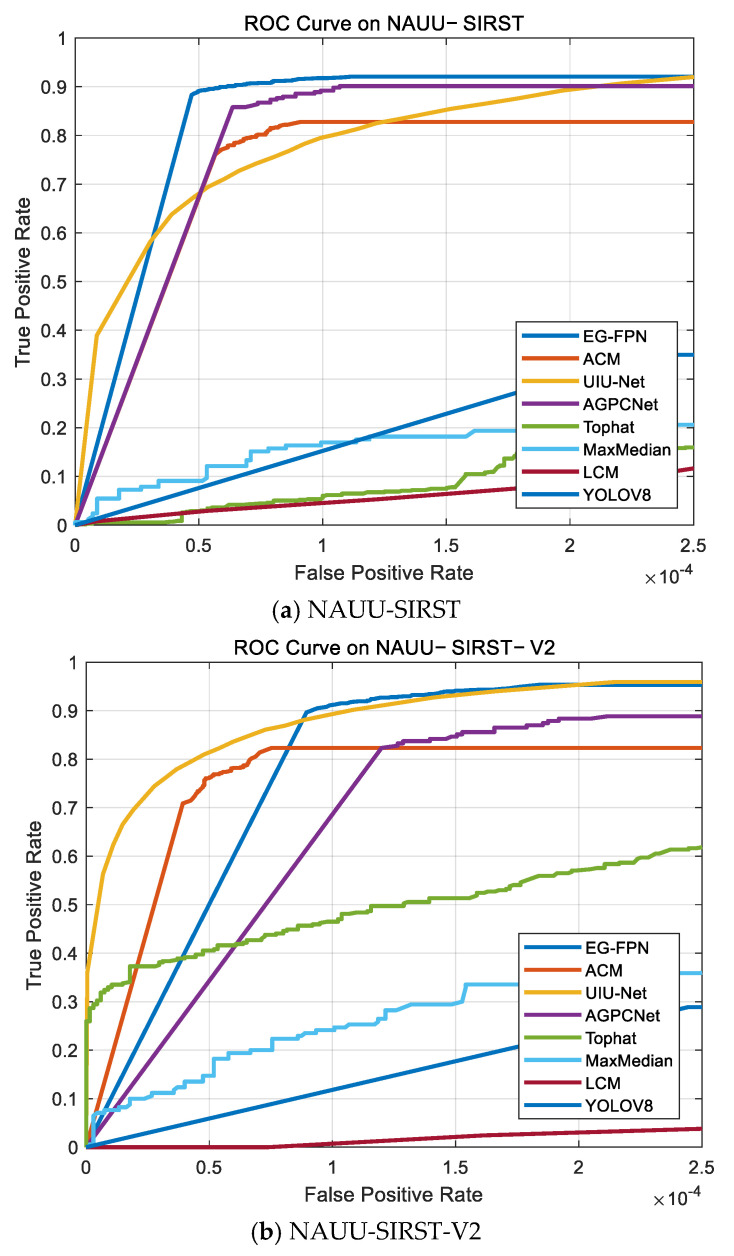
ROC curves of different methods on different datasets. (**a**) NAUU-SIRST; (**b**) NAUU-SIRST-V2 (**c**) MSISTD; (**d**) MDvsFA_cGAN.

**Figure 7 sensors-24-07767-f007:**
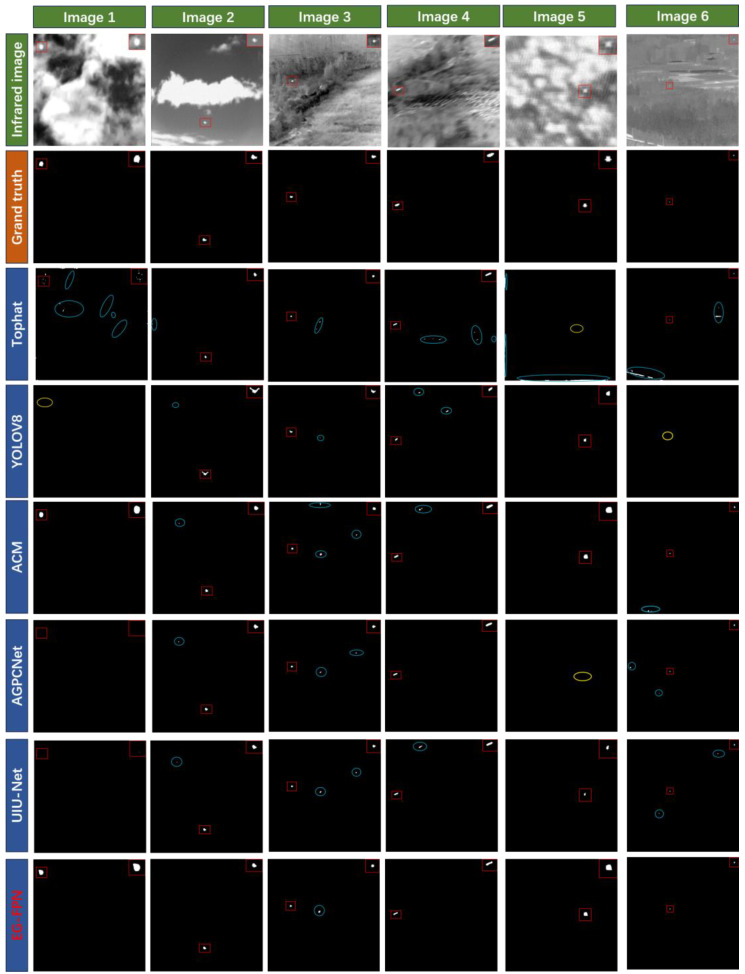
Visual comparison of various methods on the NAUU-SIRST and MSISTD datasets.

**Figure 8 sensors-24-07767-f008:**
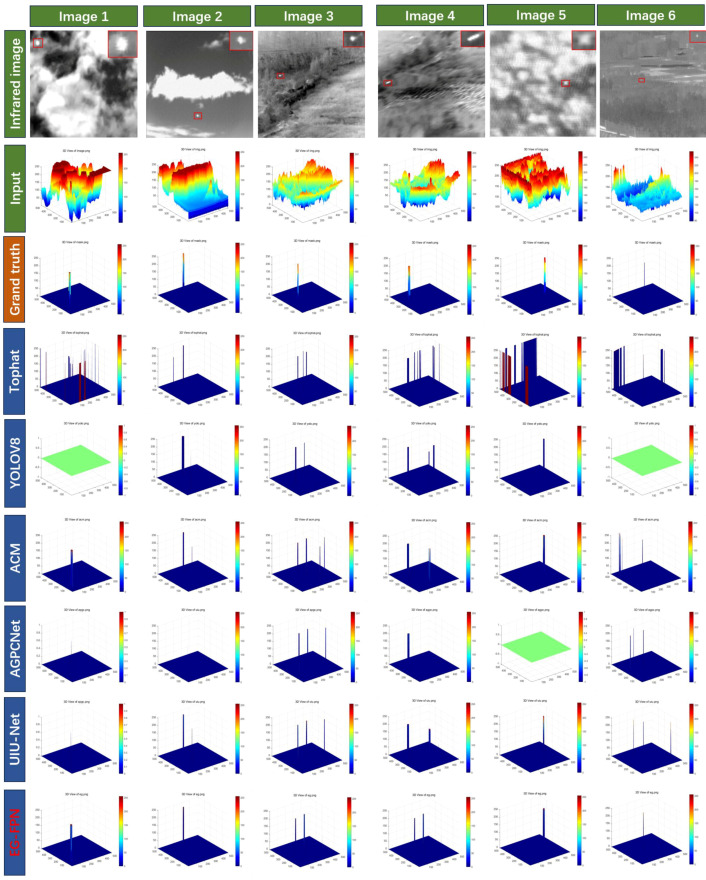
Three-dimensional visual comparison of various methods on the NAUU-SIRST and MSISTD datasets.

**Table 1 sensors-24-07767-t001:** The size of the image in each stage.

Stage	Output
Conv	120 × 120
Stage1	120 × 120
Stage2	60 × 60
Stage3	30 × 30
Fusion1	60 × 60
Fusion2	120 × 120
Predict	480 × 480

**Table 2 sensors-24-07767-t002:** Parameter settings of different datasets.

Dataset	Epochs	BatchSize	Lr	Trainval	Test
NAUU-SIRST	1000	16	0.005	341	86
NAUU-SIRST-V2	1000	16	0.005	411	103
MSISTD	500	32	0.005	900	177
MDvsFA_cGAN	500	16	0.005	801	200

**Table 3 sensors-24-07767-t003:** The detection effects of different methods.

	NAUU-SIRST			NAUU-SIRST-V2			MSISTD			MDvsFA_cGAN		
	IoU	nIoU	F1	IoU	nIoU	F1	IoU	nIoU	F1	IoU	nIoU	F1
Tophat	0.3923	0.4848	0.5829	0.3814	0.4231	0.5170	0.1529	0.2523	0.3215	0.1110	0.1386	0.1886
MaxMedian	0.1811	0.1963	0.2959	0.1702	0.1782	0.2633	0.1274	0.1213	0.2112	0.0510	0.0727	0.1205
LCM	0.0631	0.0661	0.1321	0.0508	0.0523	0.1045	0.0653	0.0657	0.1312	0.0211	0.0216	0.0419
YOLOv8	0.2321	0.2198	0.3603	0.2076	0.1919	0.3220	0.2933	0.2962	0.4571	0.1578	0.1566	0.2708
ACM	0.6846	0.6936	0.8128	0.6835	0.6383	0.8120	**0.6694**	0.6398	0.8020	0.4369	0.4267	0.6081
AGPCNet	**0.7116**	**0.7216**	0.8331	0.7012	0.6604	**0.8149**	0.6655	**0.6497**	0.8361	**0.4575**	0.4551	**0.7597**
UIU-Net	0.6778	0.7050	**0.8679**	**0.7355**	**0.7021**	0.8125	0.6596	0.6395	**0.8679**	0.4449	**0.4977**	0.6068
EG-FPN	**0.7373**	**0.7410**	**0.9086**	**0.7368**	**0.6843**	**0.8640**	**0.6711**	**0.6506**	**0.9000**	**0.4787**	**0.4924**	**0.8037**

**Table 4 sensors-24-07767-t004:** The ablation experiment results for the EG-FPN model.

FPN				IoU	nIoU	F1
HWD	APF	EAR	HMF			
×	×	×	×	0.6454	0.6198	0.8014
√	×	×	×	0.6590	0.6399	0.8050
√	√	×	×	0.6633	0.6437	0.8667
√	√	×	√	0.6664	0.6554	0.8166
√	√	√	√	0.6711	0.6506	0.9000

**Table 5 sensors-24-07767-t005:** Comparison of the performances of the different methods.

	GFLOPs	FPS
ACM	2.2393	242.2283
AGPCNet	344.7097	14.9465
UIU-Net	435.4078	33.1294
EG-FPN	1.6923	119.1858

**Table 6 sensors-24-07767-t006:** Comparison of the performances of the different methods.

Module	GFLOPs	FPS
APF	0.2336	1249.4799
HMF	0.1369	480.8322
EAR	0.0674	3593.9677

## Data Availability

Data sharing is not applicable to this article as no datasets were generated during the current study.

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
