# Peer review of "Edge-Guided Feature Pyramid Networks: An Edge-Guided Model for Enhanced Small Target Detection"

_sensors, 2024, doi:10.3390/s24237767_

Round 1

Reviewer 1 Report

Comments and Suggestions for Authors

In this work, authors manuscript introduces a novel deep learning model, EG-FPN, designed to enhance small target detection in infrared imaging, with a focus on defense applications. The model integrates edge information with multi-scale feature fusion, addressing challenges such as the loss of critical target details during feature extraction. EG-FPN employs three key components: the APF Block for multi-scale feature extraction with inter-channel attention, the HMF Block for dynamic feature weight adjustment, and the EAR Block to combine semantic and edge information. Experimental results demonstrate that EG-FPN outperforms existing state-of-the-art methods across multiple datasets, achieving superior performance metrics such as IoU, nIoU, and F1 scores. Additionally, the model exhibits lower computational complexity, with the lowest GFLOPs among tested models and competitive real-time processing capability (high FPS). The manuscript highlights EG-FPN’s effectiveness in accurately detecting and localizing small targets, even in complex and noisy infrared scenarios, making it suitable for resource-constrained applications. Please check my following comments:

 1) Expand on how the datasets (NAUU-SIRST, MSISTD, etc.) were preprocessed and divided. The details provided are brief and could confuse readers unfamiliar with these datasets.

2) Provide justification for choosing specific evaluation metrics (IoU, nIoU, F1) over others commonly used in similar studies.

3) Improve visual representation by clearly marking false positives and false negatives in the comparative visual results (Figures 7 and 8).

4) Add a detailed discussion about the robustness of EG-FPN under different weather and lighting conditions, as mentioned in the datasets section

5) Discuss the contribution of individual modules (HWD, APF, EAR, HMF) beyond numerical improvements. Highlight specific challenges each module addresses.

6) Extend comparisons with more advanced detection methods like YOLO or Transformer-based architectures to strengthen claims of superiority.

7) Discuss why specific baseline methods underperformed in detail and what aspects EG-FPN improves upon.

8) Provide a breakdown of the model's latency (FPS) and complexity (GFLOPs) per module to validate claims of efficiency.

9)  Include results from testing on resource-constrained devices if the model is proposed for real-time applications.

Author Response

Comment 1) Expand on how the datasets (NAUU-SIRST, MSISTD, etc.) were preprocessed and divided. The details provided are brief and could confuse readers unfamiliar with these datasets.
Reply 1) Thank you for pointing this out. I have added a detailed description of the dataset partitioning method from lines 302 to 313.

Comment 2)  Provide justification for choosing specific evaluation metrics (IoU, nIoU, F1) over others commonly used in similar studies.
Reply 2) Thank you for pointing this out. I have added the reasoning for choosing specific evaluation metrics (IoU, nIoU, F1) instead of other commonly used metrics in similar studies, from lines 288 to 297.

Comment 3) Improve visual representation by clearly marking false positives and false negatives in the comparative visual results (Figures 7 and 8).
Reply 3) TThank you for pointing this out. I have re-labeled the false positives and false negatives in Figure 7, and provided the corresponding explanation in lines 350 to 353. However, Figure 8 presents a 3D plot of the comparison results, which may not be ideal for labeling.

Comment 4) Add a detailed discussion about the robustness of EG-FPN under different weather and lighting conditions, as mentioned in the datasets section
Reply 4) Thank you for pointing this out. I have added a discussion on the detection performance of various models under different lighting and background conditions in lines 355 to 376.

Comment 5) Discuss the contribution of individual modules (HWD, APF, EAR, HMF) beyond numerical improvements. Highlight specific challenges each module addresses.
Reply 5) Thank you for pointing this out. I have added a discussion on the contributions of each module beyond their numerical improvements in lines 388 to 400.

Comment 6) Extend comparisons with more advanced detection methods like YOLO or Transformer-based architectures to strengthen claims of superiority.
Reply 6) Thank you for pointing this out. I have added the YOLOv8 model as a comparison method, and the comparison results are reflected in Table 3, Figure 6, Figure 7, and Figure 8.

Comment 7) Discuss why specific baseline methods underperformed in detail and what aspects EG-FPN improves upon.
Reply 7) Thank you for pointing this out. I have added a discussion, from lines 425 to 453, on the reasons for the poor performance of specific baseline methods, as well as the improvements made by EG-FPN.

Comment 8) Provide a breakdown of the model's latency (FPS) and complexity (GFLOPs) per module to validate claims of efficiency.
Reply 8) Thank you for pointing this out. In Table 6, I have provided a breakdown of the model latency (FPS) and complexity (GFLOPs) for each module, and the relevant settings are described in lines 419 to 422.

Comment 9)  Include results from testing on resource-constrained devices if the model is proposed for real-time applications.
Reply 9) Thank you for pointing this out. Due to current resource constraints, I have not been able to test the model on a wider range of devices. However, I believe the results presented in Table 5 already provide a strong indication of the model's efficiency, which can serve as a meaningful reference for its performance in real-time applications.

Reviewer 2 Report

Comments and Suggestions for Authors

This study introduces a novel model that integrates edge characteristics with multi-scale feature fusion, which aims to enhance the detection accuracy for infrared small object. The manuscript is generally well-organized, with clear sections that logically progress, and the writing is clear and technical. But before further consideration for publication, some points have to be revised:

1.       Section 2.1 related work is recommended to appear in introduction;

2.       In section 2.1, there are two key limits emphasized: real-time detection and complex background, the first is generally supported by results shown in section 3.5 but the second is less supported.

3.       Section 4 discussion is too short and lack of useful information, the authors could discuss the advantages of EG-FPN on, for example, complex background scenarios, and limitations;

4.       I guess Figure 4 has lower dpi than 300, high-resolution figure is recommended;

5.       Line 309, the long name of EAR should appear in the title of figure 1 and line 224 at the first time;

6.       Line 323, why edge information helps in small target detection?

7.       Line 356-358, the equations of precision and recall are better displayed in a single line like TPR and FPR (better separate them into two lines).

Author Response

Comment 1) Section 2.1 related work is recommended to appear in introduction.
Reply 1) Thank you for pointing this out. I have streamlined the content of Section 2.1 into the introduction, with the relevant text now located in lines 53-62, 75-79, 84-87, and 92-97.

Comment 2) In section 2.1, there are two key limits emphasized: real-time detection and complex background, the first is generally supported by results shown in section 3.5 but the second is less supported.
Reply 2) Thank you for pointing this out. I have added a discussion on the detection results under complex background conditions, from lines 355 to 376. I believe the results demonstrate that EG-FPN performs well in such challenging conditions.

Comment 3) Section 4 discussion is too short and lack of useful information, the authors could discuss the advantages of EG-FPN on, for example, complex background scenarios, and limitations;
Reply 3) Thank you for pointing this out. I have added a detailed discussion on the reasons for the poor performance of specific baseline methods and the improvements made by EG-FPN in lines 425 to 453. Additionally, I have included a discussion on the advantages of EG-FPN in lines 463 to 467.

Comment 4)  I guess Figure 4 has lower dpi than 300, high-resolution figure is recommended;
Reply 4) Thank you for pointing this out. I have replaced the resources in Figure 4.

Comment 5)  Line 309, the long name of EAR should appear in the title of figure 1 and line 224 at the first time;
Reply 5) Thank you for pointing this out. The full name of EAR has been updated to appear for the first time in line 141.

Comment 6) Line 323, why edge information helps in small target detection?
Reply 6) Thank you for pointing this out. I have added an explanation on how edge information contributes to small object detection in lines 236 to 239.

Comment 7)  Line 356-358, the equations of precision and recall are better displayed in a single line like TPR and FPR (better separate them into two lines).
Reply 7) Thank you for pointing this out. The formulation has been revised, and the changes are reflected in lines 279-280.